# A New MPPT-Based Extended Grey Wolf Optimizer for Stand-Alone PV System: A Performance Evaluation versus Four Smart MPPT Techniques in Diverse Scenarios

Mohammed Yousri Silaa [1,2], Oscar Barambones [2,*], Aissa Bencherif [1] and Abdellah Rahmani [3]

1  Telecommunications Signals and Systems Laboratory (TSS), Amar Telidji University of Laghouat, BP 37G, Laghouat 03000, Algeria; moh.silaa@lagh-univ.dz (M.Y.S.); a.bencherif@lagh-univ.dz (A.B.)
2  Engineering School of Vitoria, University of the Basque Country UPV/EHU, Nieves Cano 12, 1006 Vitoria, Spain
3  Laboratory of Physico-Chemistry of Materials (LPCM), University of Laghouat, BP 37G, Laghouat 03000, Algeria; abdellah.rahmani@lagh-univ.dz
*  Correspondence: oscar.barambones@ehu.eus

**Abstract:** Photovoltaic (PV) systems play a crucial role in clean energy systems. Effective maximum power point tracking (MPPT) techniques are essential to optimize their performance. However, conventional MPPT methods exhibit limitations and challenges in real-world scenarios characterized by rapidly changing environmental factors and various operating conditions. To address these challenges, this paper presents a performance evaluation of a novel extended grey wolf optimizer (EGWO). The EGWO has been meticulously designed in order to improve the efficiency of PV systems by rapidly tracking and maintaining the maximum power point (MPP). In this study, a comparison is made between the EGWO and other prominent MPPT techniques, including the grey wolf optimizer (GWO), equilibrium optimization algorithm (EOA), particle swarm optimization (PSO) and sin cos algorithm (SCA) techniques. To evaluate these MPPT methods, a model of a PV module integrated with a DC/DC boost converter is employed, and simulations are conducted using Simulink-MATLAB software under standard test conditions (STC) and various environmental conditions. In particular, the results demonstrate that the novel EGWO outperforms the GWO, EOA, PSO and SCA techniques and shows fast tracking speed, superior dynamic response, high robustness and minimal power fluctuations across both STC and variable conditions. Thus, a power fluctuation of 0.09 W could be achieved by using the proposed EGWO technique. Finally, according to these results, the proposed approach can offer an improvement in energy consumption. These findings underscore the potential benefits of employing the novel MPPT EGWO to enhance the efficiency and performance of MPPT in PV systems. Further exploration of this intelligent technique could lead to significant advancements in optimizing PV system performance, making it a promising option for real-world applications.

**Keywords:** PV system; MPPT; EGWO; GWO; EOA; PSO; SCA

## 1. Introduction

### 1.1. Motivations

Solar energy has emerged as one of the most promising renewable energy sources, offering sustainable and eco-friendly solutions to meet the ever-growing global energy demand over conventional fossil fuel-based sources [1]. Among the various solar technologies, photovoltaic (PV) systems have garnered significant attention due to their direct conversion of sunlight into electricity [2]. As concerns about climate change and environmental sustainability intensify, the adoption of PV systems continues to grow in various sectors [3]. PV systems are widely deployed in residential, commercial and industrial applications, as well as in smart microgrids, contributing to a cleaner and greener future [4,5]. However, the efficiency of PV systems is strongly influenced by dynamic environmental conditions,

such as varying solar irradiance and temperature [6]. The maximum power point tracking (MPPT) technique plays a pivotal role in optimizing the energy extraction from PV modules under fluctuating conditions [7]. In particular, those techniques enable the PV system to operate at its maximum power output, ensuring enhanced energy efficiency and increased power generation [8]. One of the most common devices to integrate MPPT into a PV system is a DC/DC converter. The boost converter is a type of DC/DC converter that raises the voltage from the solar panels to a higher level, making it suitable for charging batteries or feeding power into the grid [9]. The combination of MPPT and a boost converter allows for real-time tracking of the MPP and an efficient power conversion ratio, optimizing the system's energy output and increasing its overall cost-effectiveness [10].

### 1.2. State of the Art

As the demand for renewable energy sources continues to increase, researchers and engineers have been actively working to improve the output power of PV systems by refining MPPT algorithms and other control techniques. The following describes the research progress in recent years. Conventional MPPT methods, such as perturb and observe (P&O) [11], incremental conductance (INC) [12], fractional open circuit voltage (FOCV) [13], fractional short circuit current (FSCC) [14] and hill climbing (HL) [15], have been widely studied and adopted due to their simplicity and effectiveness. These methods employ algorithms that continuously track and adjust the operating point of the PV system to maintain it at the MPP, thus optimizing its energy conversion efficiency. Although conventional MPPT techniques have shown satisfactory performance under standard conditions, they face challenges in real-world scenarios with rapidly changing environmental factors, such as solar irradiation, temperature and shading [16]. As a result, researchers and engineers have focused on improving the adaptability, accuracy and robustness of MPPT algorithms to ensure consistent performance under varying conditions. One of the most popular classes of optimization algorithms used by researchers is swarm intelligence (SI) or smart techniques. Those algorithms draw inspiration from the social behavior of various species and have shown promising results in finding the MPP in solar PV systems. Some notable SI algorithms include particle swarm optimization (PSO) [17], artificial bee colony algorithm (ABC) [18], genetic algorithm (GA) [19], ant colony optimization (ACO) [20], firefly algorithm (FA) [21], grey wolf optimizer (GWO) [22], whale optimization algorithm (WOA) [23], cuckoo search (CS) algorithm [24], artificial fish swarm algorithm (AFSA) [25] and so on. These swarm MPPT techniques leverage the collective behavior and self-organization of agents in a swarm to efficiently explore the solution space and find the optimal solution presented in the operating point of PV systems [26]. Their ability to adapt to changing environmental conditions and global optimization make them promising tools for improving the efficiency of solar energy harvesting. Among the existing research, Mohanty et al. [27] presented a comparative study evaluating the performance of different MPPT techniques currently used in solar PV systems. Their work provides valuable insights into the strengths and weaknesses of these techniques, aiding in the selection of the most appropriate MPPT method for specific application scenarios. Calvinho et al. [28] proposed a PSO MPPT technique with a variable step size in order to reduce unwanted power oscillations. Hence, the results showed that the proposed technique effectively reduces the power oscillation around the MPP. Rajkumar et al. [29] implemented a GWO MPPT-controlled DC/DC converter linked to a PV system. The proposed approach effectively addresses certain drawbacks, such as reduced tracking efficiency, sustained oscillations at the steady state and transient issues typically encountered in the P&O and PSO methods. Based on the results obtained from simulations and experimental tests, it is evident that the proposed MPPT algorithm outperforms both P&O and PSO-based MPPT systems. Soufyane et al. [30] designed a new MPPT-based ABC optimization technique. The innovative algorithm not only overcomes the limitations commonly associated with traditional MPPT methods but also provides a straightforward and robust MPPT solution. The effectiveness of this method is verified using a co-simulation approach, combining Matlab/Simulink and Ca-

dence/Pspice, to compare its performance with the PSO-based MPPT algorithm under dynamic weather conditions. Additionally, experimental validation is conducted using a laboratory setup. Both simulation and experimental results confirm the effectiveness of the proposed approach. Compared to the PSO-based MPPT algorithm, the new ABC demonstrated superior tracking performance in locating the global MPP, especially under partially shaded and dynamic weather conditions. Notably, the ABC is not sensitive to initial conditions and does not require knowledge of the PV array characteristics. Furthermore, experimental results affirm its ability to accurately track the MPP of the PV array under partial shading conditions. Priyadarshi et al. [31] implemented a hybrid solar–wind standalone power system equipped with an MPPT to generate electrical power for residential applications in rural areas. To efficiently harness power from the wind energy system, an ACO algorithm is employed. In contrast to classical proportional-integral (PI) control, the study adopts a fuzzy logic control (FLC) inverter control strategy. The MPPT functionality is executed using a single cuk converter acting as an impedance power adapter, eliminating the need for additional voltage and current circuits, thereby enhancing the conversion efficiency of the converter and maximizing power output stages. According to the results, the proposed ACO facilitates rapid battery charging and efficient power distribution within the hybrid PV–wind system. Notably, the ACO demonstrated a sevenfold faster convergence rate compared to the PSO technique in achieving the MPP and tracking efficiency. Ahmed et al. [32] outlined the concept of CS MPPT by highlighting the significance of Lévy flight in influencing the algorithm's convergence. This, finally, explained the main equations that govern the behavior of the search. To justify CS as a viable MPPT option, a comprehensive assessment was carried out against two well-established methods, P&O and PSO. The evaluations included: gradual irradiance and temperature changes, step changes in irradiance, and finally, rapid changes in both irradiance and temperature. These tests were conducted for both large and medium-sized PV systems. Furthermore, the ability of the algorithm to handle the partial shading condition was demonstrated. The results showed that CS was capable of tracking MPP within 100–250 ms under various types of environmental changes. Additionally, the power loss in steady state due to MPP mismatch was only 0.000008%. Furthermore, it could handle the partial shading condition very efficiently. As a result, CS outperformed both P&O and PSO with respect to tracking capability, transient behavior and convergence. However, some MPPT systems combine multiple algorithms or employ adaptive techniques to switch between algorithms based on specific conditions. Hybrid methods can optimize the MPPT performance under different weather and environmental conditions. In the pursuit of further improving the performance of conventional MPPT techniques, researchers have explored innovative hybrid approaches. For instance, an improved P&O algorithm, ABC, was proposed by [33], which combines the simplicity of the P&O method and the intelligent search capabilities of the ABC algorithm. This integration results have faster convergence and increased accuracy in tracking the MPP, overcoming some of the limitations of the traditional P&O technique. Figueiredo et al. [34] introduced a hybrid P&O-PSO MPPT technique and evaluated its performance against traditional methods, including P&O and the standard PSO. Simulation outcomes demonstrated that the proposed hybrid algorithm excelled in tracking the global MPP under both uniform and partial shading conditions, with a tracking time 50% shorter than the standard PSO technique. Additionally, the proposed method extracted 0.3% more power from the photovoltaic system compared to the P&O-PSO hybrid approach, highlighting its effectiveness in improving energy generation. Chao et al. [35] combined two sequential convex methods (SCMs), GA and ACO, in order to enhance the robustness and speed of the MPPT technique. In the simulation conducted using Matlab, a GA-ACO MPPT controller was employed, utilizing four SunPower SPR-305NE-WHT-D PV modules connected in series, each with a maximum power rating of 305.226 W. These tests were performed under partial shade conditions to assess the performance of the newly proposed MPPT controller. The results were subsequently analyzed and compared with those obtained from P&O MPPT and conventional ACO MPPT techniques. The proposed GA-ACO emerged as the

swiftest performer, occasionally approaching the global MPP as early as the first iteration. In contrast, both the P&O MPPT and ACO MPPT algorithms required over 20 iterations to find a solution, and they often fell short of reaching the global MPP altogether. Additionally, the GA-ACO achieved its objectives in just 10 iterations, sometimes even fewer. This translated to a speed advantage of at least 50% over the other two algorithms. Furthermore, the GA-ACO exhibited accuracy, stability and robustness, consistently reaching the global MPP quickly, even under challenging conditions. In line with the quest for enhanced performance and adaptability in renewable energy systems, the optimization techniques mentioned above have showcased their potential to bolster the effectiveness of PV systems and microgrids. These advancements have primarily centered around improving control strategies, robustness and frequency stability. However, an equally crucial aspect that demands attention is the scalability of these methods to accommodate diverse system sizes and configurations, given the growing prevalence of large-scale PV installations and high-res renewable energy penetration in microgrids. The ability to seamlessly adapt to varying scales is a fundamental requirement for these techniques to be effective in real-world applications, from small residential PV arrays to expansive utility-scale solar farms and complex microgrid networks. Hence, Kerdphol et al. [36] introduced H∞ robust control to enhance frequency stability in high-RES-penetration islanded microgrids, overcoming issues of weakened system inertia. The comparative analysis demonstrates superior frequency tracking and disturbance attenuation, making the H∞-based virtual inertia controller a robust solution for such microgrids. Carli et al. [37] addressed an energy scheduling in a network of users sharing a renewable energy source. It combines social welfare optimization for energy allocation and cost optimization for user appliances under time-varying pricing. A decentralized optimization algorithm, using Gauss–Seidel decomposition and competitive games is proposed. Case studies in various scenarios show that this approach leverages renewable energy sharing to reduce individual costs, manage peak loads and meet customer energy needs effectively.

### 1.3. Contributions

This paper focuses on implementing an innovative extended grey wolf optimizer (EGWO) MPPT for PV systems. Hence, the EGWO is employed to address the control problem that conventional population-based algorithms face due to their limited ability to efficiently handle uncertainty and nonlinear systems, as well as their inflexibility in adapting to changing environmental conditions. The objective is to explore and validate the effectiveness of the proposed technique in order to improve the performance of PV systems. Hence, a comparative study was conducted under four techniques: the conventional grey wolf optimizer (GWO), particle swarm optimization (PSO), equilibrium optimization algorithm (EOA) and sin cos algorithm (SCA). The proposed EGWO-based MPPT algorithm is designed to dynamically adjust the duty cycle of the DC/DC boost converter, regulating the voltage and current to achieve the MPP of the PV system. By optimizing the duty cycle in real-time, the PV system can continuously track the MPP, irrespective of changes in environmental conditions, leading to improved energy harvesting efficiency. Through comprehensive simulations, the aim was to compare the performance of the MPPT techniques in terms of efficiency and minimal power fluctuations. The comparative results shed light on the superior capabilities of the EGWO algorithm in achieving higher energy conversion efficiency and stability under dynamic operating conditions.

### 1.4. Structure Overview

This paper is divided into three sections. Section 2 describes the PV system plant modeling. Section 3 is devoted to the MPPT control technique. Section 4 presents simulation results and the conclusions.

## 2. PV System Plant Modeling

*2.1. PV Mathematical Model*

In the context of PV systems, a mathematical model is used in order to describe the behavior and performance of a PV module or array under varying environmental conditions. This model helps predict the electrical output of the PV system based on factors, such as solar irradiance, temperature and the characteristics of the PV components. The fundamental equation governing the output current $I_{out}$ of a PV system is given by the following Equation [38]:

$$I_{out} = I_{ph} - I_d - I_{sh} \tag{1}$$

where $I_{sh}$ refers to the shunt or parallel resistance current, which is given by the following [38]:

$$I_{sh} = \frac{V + I \cdot R_s}{R_{sh}} \tag{2}$$

where $V$ is the voltage across the PV cell, $I$ is the current of the PV cell (output current), $R_s$ is the resistance of the cell in series and $R_{sh}$ is the resistance of the shunt.

The current generated by light ($I_{ph}$) in a photovoltaic cell is a critical parameter that represents the amount of current produced by the cell when exposed to light. It is directly proportional to the incident light intensity on the solar cell. The mathematical equation to calculate $I_{ph}$ is given as follows [39]:

$$I_{ph} = [I_{sc} + K_I(T_c - T_r)] \times \frac{G}{G_{STC}} \tag{3}$$

where $I_{sc}$ represents the short-circuit current, which is the current produced by the photovoltaic cell when its terminals are short-circuited, and the voltage across the terminals is zero, $K_I$ is the temperature coefficient of the short-circuit current. It describes how the short-circuit current varies with changes in temperature, $T_c$ refers to the cell temperature, $T_r$ denotes the reference temperature in degrees (°C), which is a standard temperature used as a reference point for calculating the temperature-dependent parameters, $G$ and $G_{STC}$ represents the actual irradiance or light intensity falling on the photovoltaic cell and the standard test condition irradiance, which is a reference irradiance value used for standardizing photovoltaic cell performance measurements, respectively (W/m²). The diode current $I_d$ in a PV system can be calculated using the diode equation, which describes the current–voltage characteristic of a diode. The diode current can be expressed as follows [40]:

$$I_d = I_s \left( e^{\frac{q \cdot V_d}{n \cdot K T_c}} - 1 \right) \tag{4}$$

where $I_s$ is the reverse saturation current of the diode, $q$ is the electron charge (coulombs (C)), $n$ is the diode ideality factor, $K$ is the Boltzmann constant (m²·kg·s⁻²·K⁻¹) and $V_d$ is the voltage of the equivalent diode, which is calculated as follows [41]:

$$V_d = V + I \cdot R_s \tag{5}$$

Finally, the total current generated by the PV cell $I_{pv}$ is given by [41]:

$$I_{pv} = I_{ph} - I_0 \left( \exp\left( \frac{q(V + I \cdot R_s)}{n \cdot K \cdot T_c \cdot N_s} \right) - 1 \right) \tag{6}$$

where $N_s$ is the number of series-connected cells used to adjust the current based on the cell configuration. The specification values of the PV array type (**Sun Earth Solar Power TDB156x156-60-P 215W**) under standard test conditions (STC) considered in the simulation are listed in Table 1.

**Table 1.** Photovoltaic array model parameter values.

| Specifications | Value |
|---|---|
| Maximum power (W) | 215.028 |
| Cells per module ($N_{cell}$) | 60 |
| Open circuit voltage $V_{oc}$ (V) | 36.8 |
| Short-circuit current $I_{sc}$ (A) | 7.92 |
| Voltage at maximum power point $V_{mp}$ (V) | 29.7 |
| Current at maximum power point $I_{mp}$ (A) | 7.24 |
| Temperature coefficient of $V_{oc}$ (%/deg.C) | −0.34 |
| Temperature coefficient of $I_{sc}$ (%/deg.C) | 0.05 |

*2.2. DC/DC Boost Converter*

This study focuses on the modeling and control of a DC/DC converter, specifically a step-up converter, in the context of a PV system. This type of converter has the ability to step up a lower input voltage into a higher output voltage via a controlled pulse-width-modulation (PWM) switching technique. In this context, the duty cycle (*d*) determines the average output voltage. A higher duty cycle results in a higher average voltage, and a lower duty cycle results in a lower average voltage. The PWM controllers use feedback mechanisms to adjust the duty cycle of the converter to track the MPP of the PV module under changing environmental conditions. DC/DC converters are essential components in PV systems, allowing voltage step-up or step-down operations, depending on the load requirements [42]. The boost converter duty cycle behavior is mathematically described by Equation (7) [43].

$$d = 1 - \frac{V_{in}}{V_{out}} \tag{7}$$

This work explores a closed-loop system (Figure 1) that includes the step-up converter and the integration of an MPPT with the PV system. The primary objective is to enhance the overall performance and efficiency of the PV system by accurately determining and tracking the MPP of the PV module under varying environmental conditions. Such a control approach can find applications in various large-scale systems [44–47], where the high efficiency of conversion of power is crucial [48]. The study explores the effectiveness of the proposed MPPT technique in optimizing the performance of the PV system and ensuring reliable power delivery.

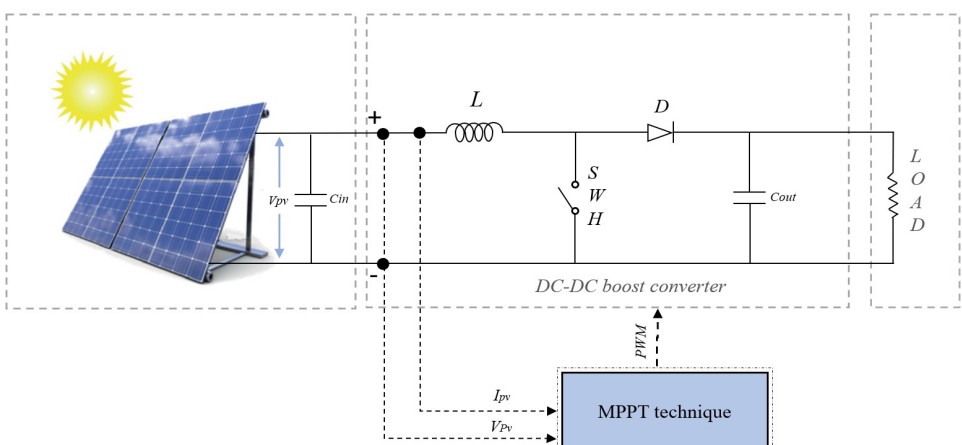

**Figure 1.** Closed-loop system.

## 3. MPPT Control Method

*3.1. Extended and Grey Wolf Optimizers*

The grey wolf optimizer (GWO) is a nature-inspired swarm intelligence algorithm that draws inspiration from the cooperative hunting behavior of grey wolves. In the

wild, wolves are known for their remarkable group dynamics and hierarchical leadership structure during hunts. The GWO algorithm models this social hierarchy to search for optimal solutions in complex optimization problems efficiently. The population of wolves in GWO is divided into four groups: alpha ($\alpha$), beta ($\beta$), delta ($\delta$) and omega ($\omega$). The $\alpha$, $\beta$ and $\delta$ wolves are considered the most influential and dominant, while the $\omega$ wolves are the weaker members guided by the top three in exploring promising regions of the solution space. By mimicking the collaboration and leadership within a wolf pack, the GWO algorithm effectively strikes a balance between exploration and exploitation, making it a powerful optimization tool [49].

3.1.1. GWO Mathematical Model

The main stages of the GWO are based on the following behaviors [50]:

- Social Hierarchy: The social hierarchy of the wolves is represented by four positions: $X_\alpha$, $X_\beta$, $X_\delta$ and $X_\omega$. These positions represent the best, second-best, third-best and the rest of the wolves in the population, respectively.
- Encircling behavior: Entails the coordinated movement of group members toward a specific target position while tightening the search area around it. The wolves concentrate their exploration around influential leaders ($\alpha$, $\beta$, $\delta$), optimizing the balance between exploration and exploitation. This approach enables the algorithm to effectively discover optimal or near-optimal solutions for complex optimization problems. The encircling behavior equations are given as follows [49]:

$$\overrightarrow{D} = |\overrightarrow{C} \cdot \overrightarrow{X_p}(t) - \overrightarrow{X}(t)| \tag{8}$$

$$\overrightarrow{X}(t+1) = |\overrightarrow{X_p}(t) - \overrightarrow{A} \cdot \overrightarrow{D}| \tag{9}$$

where $\overrightarrow{X}(t+1)$ is the updated position of the $i - th$ wolf at time step $t + 1$, $\overrightarrow{X}(t)$ is the current position of the wolf, $\overrightarrow{X_p}$ is the position of the targeted wolf, $\overrightarrow{A}$ and $\overrightarrow{C}$ are random coefficients vectors that determine the encircling behavior which lies in the range $[-1, 1]$. The equation to generate the $\overrightarrow{A}$ and $\overrightarrow{C}$ vectors are given as follows [49]:

$$\overrightarrow{A} = 2\overrightarrow{a} \cdot \overrightarrow{r}_1 - \overrightarrow{a} \tag{10}$$

$$\overrightarrow{C} = 2 \cdot \overrightarrow{r}_2 \tag{11}$$

where $\overrightarrow{a}$ is a random coefficient vector with elements uniformly decreased from 2 to 0, and $\overrightarrow{r}_1$, $\overrightarrow{r}_2$ are random coefficient vectors with elements uniformly distributed in $[-1,1]$.

- Follow, hunt and approach the prey: The $\alpha$, $\beta$, $\delta$ wolves guide the $\omega$ wolves toward promising regions. The updated position for each omega wolf is determined by the influence of the $\alpha$, $\beta$ and $\delta$ wolves as follows:

$$\overrightarrow{D_\alpha} = |\overrightarrow{C_1}\overrightarrow{X_\alpha}(t) - \overrightarrow{X}(t)|, \overrightarrow{X}_1 = \overrightarrow{X_\alpha} - \overrightarrow{A_1} \cdot \overrightarrow{D_\alpha} \tag{12}$$

$$\overrightarrow{D_\beta} = |\overrightarrow{C_2}\overrightarrow{X_\beta}(t) - \overrightarrow{X}(t)|, \overrightarrow{X}_2 = \overrightarrow{X_\beta} - \overrightarrow{A_2} \cdot \overrightarrow{D_\beta} \tag{13}$$

$$\overrightarrow{D_\delta} = |\overrightarrow{C_3}\overrightarrow{X_\delta}(t) - \overrightarrow{X}(t)|, \overrightarrow{X}_3 = \overrightarrow{X_\delta} - \overrightarrow{A_3} \cdot \overrightarrow{D_\delta} \tag{14}$$

Therefore, the updated position of all search agents is given by [49]:

$$\overrightarrow{X}(t+1) = \frac{\overrightarrow{X_1} + \overrightarrow{X_2} + \overrightarrow{X_3}}{3} \tag{15}$$

### 3.1.2. EGWO Mathematical Model

In contrast, the EGWO introduces an array of enhancements to augment the algorithm's exploration–exploitation balance and convergence speed. These include the incorporation of three dynamic coefficients to control the exploration rate, a finer initialization of the population for increased diversity and the utilization of the mean position of the population for solution updates. Such modifications empower EGWO with enhanced adaptability, making it particularly well-suited for complex optimization problems requiring swift convergence and robust global search. The updated position of all search agents using EGWO is given as follows [51]:

$$\overrightarrow{X}(t+1) = \frac{\alpha_{em}\overrightarrow{X_1} + \beta_{em}\overrightarrow{X_2} + \delta_{em}\overrightarrow{X_3}}{3} \tag{16}$$

where $\alpha_{em} > \beta_{em} > \delta_{em}$ are called the emphasis coefficients.

### 3.1.3. EGWO and GWO Application for MPPT

The primary objective is to maximize the output power (P) from the PV array by optimizing the duty cycle (*d*), which represents the fraction of time a power converter is on. According to [52], the objective function is defined as follows:

$$d_{min} \leq d \leq d_{max} \tag{17}$$

The sequential procedure for achieving the MPP using the EGWO and GWO MPPTs algorithms unfolds as follows:

- **Initialization**: The optimization process starts with the initialization of a population $N_p$ (represented by wolves) in the search space. The duty ratio $d_i$ is initialized (Equation (18)) randomly within the defined limits, ranging from 0.1 to 0.9.

$$d_i = rand(N_p, 1)(d_{max} - d_{min}) + d_{min} \tag{18}$$

- **Evaluation**: The fitness values, corresponding to the PV power output, are calculated for each member of the population. The wolves with the highest PV power values are assigned as $d\alpha$ (the best solution), $d\beta$ (the second-best solution) and $d\delta$ (the third-best solution).

- **Updating Positions**: The positions $d_i$ (duty ratios) of the wolves in the population are updated based on the positions of $d\alpha$, $d\beta$ and $d\delta$, the best, second and the third-best solutions, respectively. This update aims to explore the search space more effectively and improve the duty ratios. The updated position of all search agents using GWO and EGWO is given as follows [52]:

$$\overrightarrow{D_\alpha} = |\overrightarrow{C_1}\overrightarrow{d_\alpha} - \overrightarrow{d_i}|, \ \overrightarrow{d}_1 = \overrightarrow{d_\alpha} - \overrightarrow{A_1} \cdot \overrightarrow{D_\alpha} \tag{19}$$

$$\overrightarrow{D_\beta} = |\overrightarrow{C_2}\overrightarrow{d_\beta} - \overrightarrow{d_i}|, \ \overrightarrow{d}_2 = \overrightarrow{d_\beta} - \overrightarrow{A_2} \cdot \overrightarrow{D_\beta} \tag{20}$$

$$\overrightarrow{D_\delta} = |\overrightarrow{C_3}\overrightarrow{d_\delta} - \overrightarrow{d_i}|, \ \overrightarrow{d}_3 = \overrightarrow{d_\delta} - \overrightarrow{A_3} \cdot \overrightarrow{D_\delta} \tag{21}$$

where $\overrightarrow{d_1}$, $\overrightarrow{d_2}$, $\overrightarrow{d_3}$ are the duty cycle direction vectors for the $\alpha$, $\beta$ and $\delta$ wolves, respectively. In the EGWO and GWO algorithms with direct duty-cycle control for MPPT in PV systems, the updated duty cycle vector $d(t + 1)$ is determined based on the power output of the wolves. Hence, using (15), (16), (19), (20) and (21), the duty cycle update equation for GWO and EGWO are given as follows, respectively:

$$\overrightarrow{d_i}(t+1) = \frac{\overrightarrow{d_1} + \overrightarrow{d_2} + \overrightarrow{d_3}}{3} \tag{22}$$

$$\vec{d_i}(t+1) = \frac{\alpha_{em}\vec{d_1} + \beta_{em}\vec{d_2} + \delta_{em}\vec{d_3}}{3} \tag{23}$$

- **Termination**: The termination condition is determined by the maximum number of iterations reached or when the relative change in PV power compared to the previous iteration's power becomes negligible. The termination criterion is defined as follows:

$$\frac{|P_{pv} - P_{pv,old}|}{P_{pv,old}} \geq \Delta p \tag{24}$$

where $P_{pv}$ represents the PV power calculated for the current duty cycle, $P_{pv,old}$ represents the PV power calculated for the previous duty cycle and $\Delta p$ is set to 10%, representing the predefined threshold for the relative change in the PV power. The EGWO MPPT flowchart is given in Figure 2.

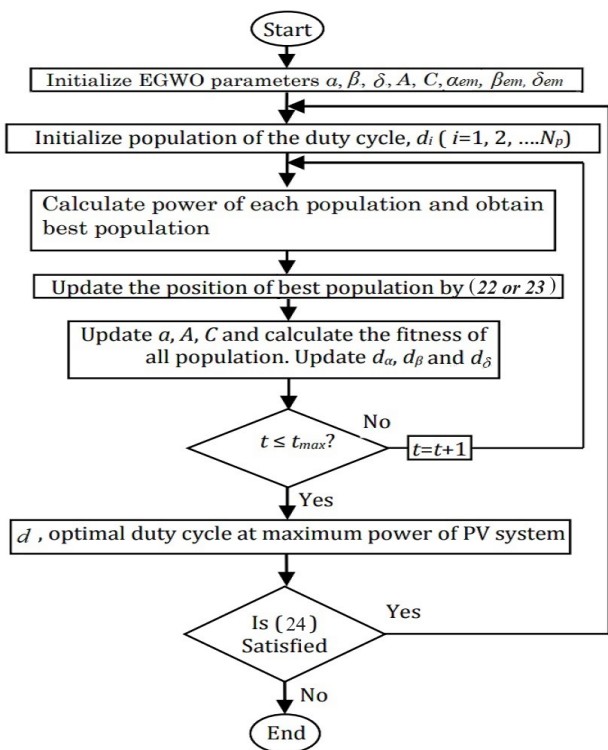

**Figure 2.** Extended grey wolf optimizer (EGWO) maximum power point tracking (MPPT) flowchart [52].

## 4. Simulation Results

This research aims to extract the maximum power from a PV system. The application involves using the EGWO algorithm for MPPT control. The primary goal is to ensure that the PV system operates at the desired MPP, extracting the highest possible power output under different scenarios applied to validate the MPPT technique robustness. The DC/DC boost converter parameters used in the simulation are summarized in Table 2.

**Table 2.** Boost converter parameters value.

| Specifications | Value |
|---|---|
| Inductance (L) | 0.15 H |
| In capacitor (C) | $100 \times 10^{-6}$ F |
| Out capacitor (C) | $470 \times 10^{-6}$ F |
| Max $f_{SW}$ | 10 kHz |
| Load | 32 Ohm |

Figure 3 illustrates the P–V curve of the PV array (**Sun Earth Solar Power TDB156x156-60-P 215W**) under different irradiance levels and temperatures. Additionally, the proposed MPPT parameters are listed in Table 3.

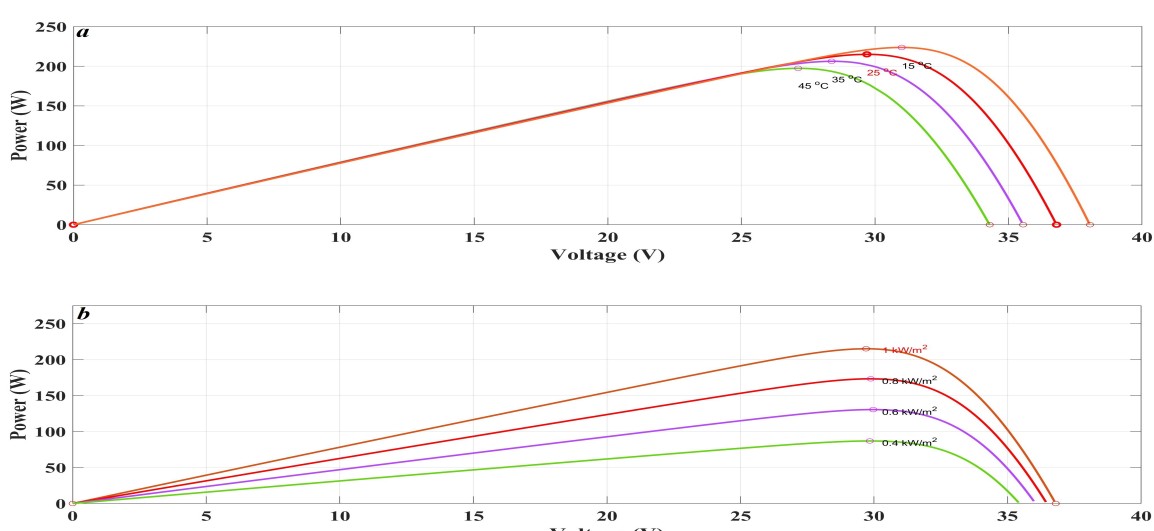

**Figure 3.** Photo–voltaic curve. (**a**): Temperature variations. (**b**): Irradiance variations.

**Table 3.** Parameters of maximum power point tracking (MPPT) algorithms.

| Parameter | EGWO | GWO | PSO | EOA | SCA |
|---|---|---|---|---|---|
| Population size ($N_p$) | 20 | 20 | 20 | 20 | 20 |
| Maximum number of iterations ($t_{max}$) | 100 | 100 | 100 | 100 | 100 |
| $A$ and $C$ | Random | Random | – | – | – |
| $\alpha_{em}$ | 1.5 | – | – | – | – |
| $\beta_{em}$ | 1.2 | – | – | – | – |
| $\delta_{em}$ | 1.1 | – | – | – | – |
| $w_{inertia}$ | – | – | 0.7 | – | – |
| $C_{cognitive}$ | – | – | 1.4 | – | – |
| $C_{social}$ | – | – | 1.4 | – | – |
| $a_1$ | – | – | – | 2 | – |
| $a_2$ | – | – | – | 1 | – |
| $GP$ | – | – | – | 0.5 | – |
| $r_{1,2,3,4}$ | – | – | – | – | Random in $[0, 1]$ |

*4.1. First Scenario: Under Standard Test Conditions (STC)*

In this condition, the solar irradiance is set to a constant value of 1000 W/m², representing the solar radiation intensity on a clear day. The cell temperature is maintained at 25 °C, which is often referred to as the module temperature and represents the typical operating temperature of PV cells during testing. Figure 4 shows the PV system power under EGWO, GWO, PSO, EOA and SCA. The results indicate that EGWO exhibits outstanding performance, demonstrating a rapid response time and minimal power fluctuations around the MPPT point, with a power oscillation (OS) of approximately 0.09 W. On the other hand,

the GWO ranks as the second-best performer among the algorithms tested, with a power oscillation of 0.266 W, highlighting its ability to maintain stable power output near the MPPT point under the specified conditions. However, it is important to note that the power oscillations, such as those observed with PSO, SCA and EOA algorithms (0.731 W, 0.729 W, and 1.044 W, respectively), can have significant consequences. These oscillations can lead to efficiency losses, decreased PV system lifespan, reduced energy yield and adverse effects on systems with energy storage.

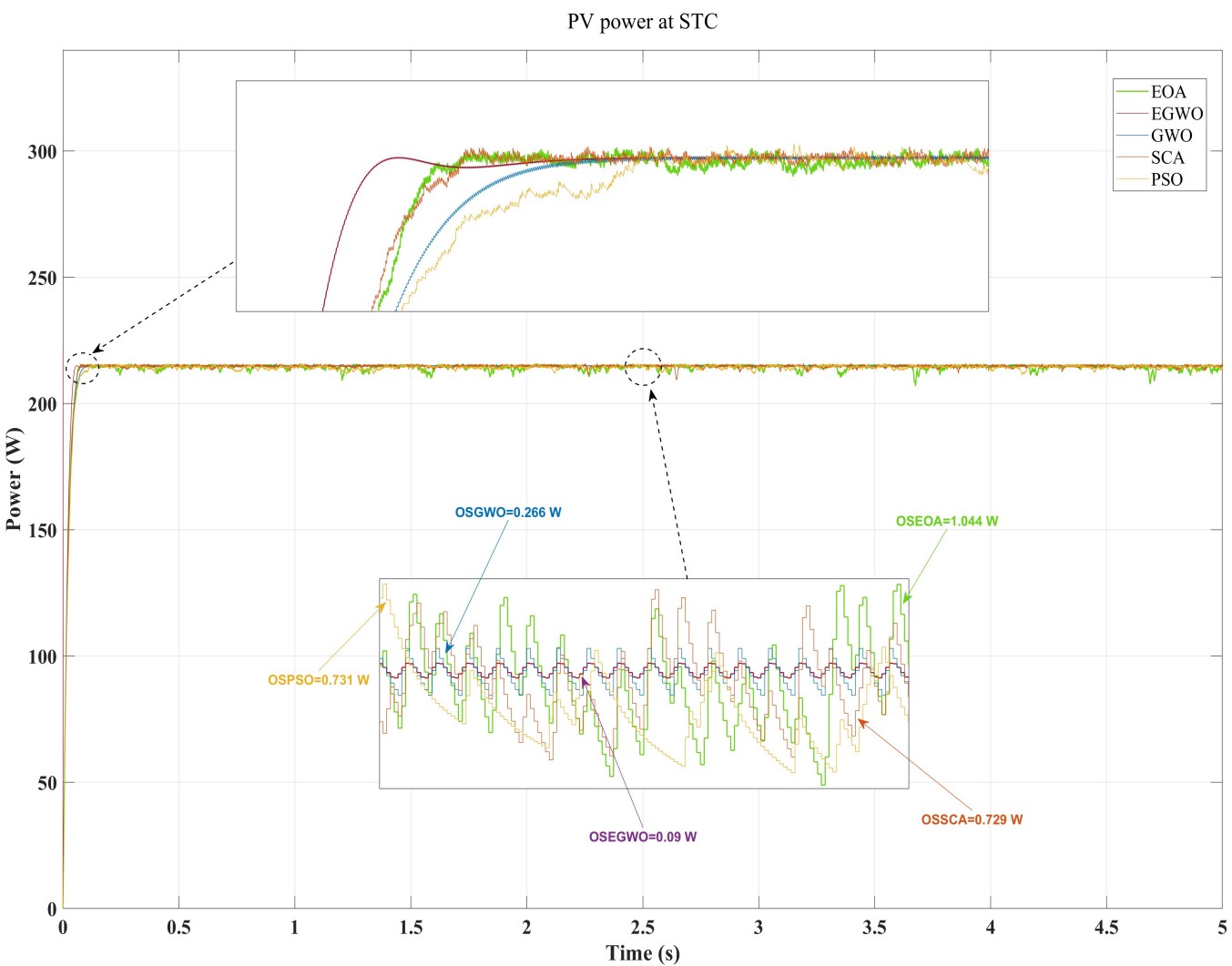

**Figure 4.** Photovoltaic power under standard test conditions.

### 4.2. Second Scenario: Variable Irradiance and Constant Temperature

Figure 5 represents the second scenario. In this scenario, variable irradiance is applied from a lower value of 850 W/m$^2$ to 1000 W/m$^2$ at a constant temperature T = 25 °C.

Figure 6 shows the PV system power under EGWO, GWO, PSO, EOA and SCA. The results indicate for the second time that the EGWO exhibits outstanding performance in terms of minimal power fluctuations around the MPPT point, with a power oscillation equal to 0.09 W. Meanwhile, at the application of the irradiance (period of 1.5 s and 2 s) the GWO, PSO, SCA and EOA algorithms show a power oscillation equal to 0.266 W, 18.401 W, 15.796 W and 26.671 W, respectively. Although, at the period of 4 s and 5 s, the EGWO and GWO keep the same performance in terms of low power osculation. On the other hand, the PSO, EOA and SCA algorithms show a power oscillation equal to 15.902 W, 35.9121 W and 33.963 W, respectively.

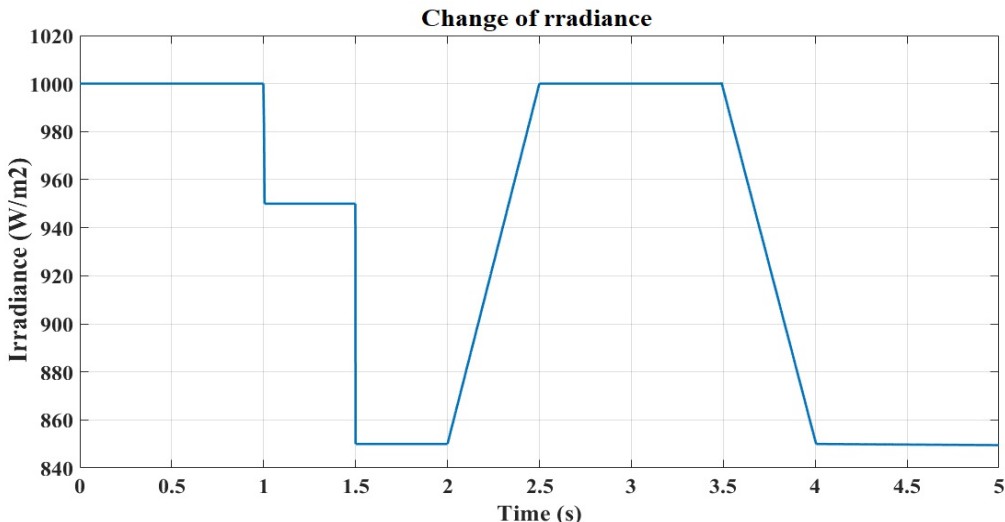

**Figure 5.** Variable irradiance at a temperature of 25 °C.

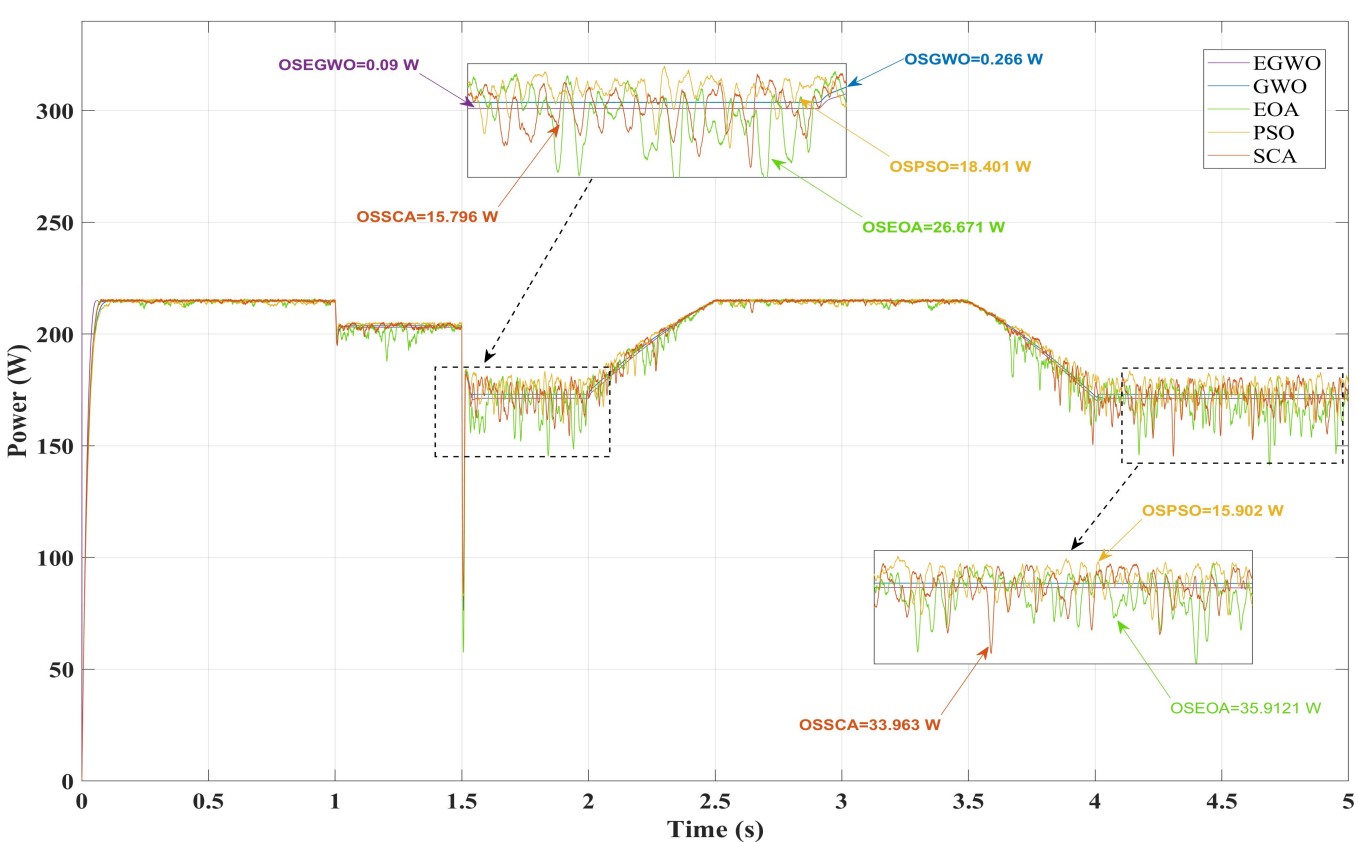

**Figure 6.** Photovoltaic power under variable irradiance and fixed temperature.

### 4.3. Third Scenario: Variable Temperature and Constant Irradiance

Figure 7 represents the third scenario. In this scenario, temperature variations are applied from a lower value of 25 °C to a higher value of 40 °C, while the irradiance is considered fixed at whole.

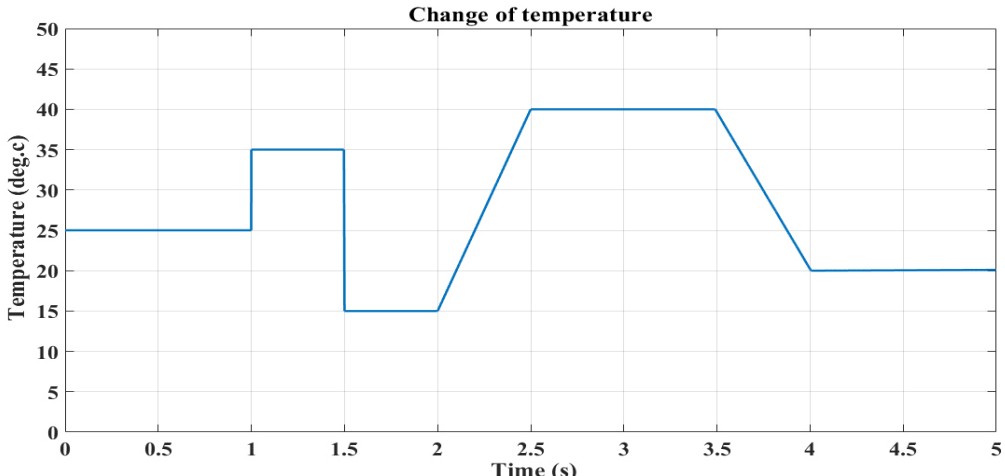

**Figure 7.** Variable temperature at actual irradiance (G) = 1000 W/m$^2$.

Figure 8 shows the PV system power under EGWO, GWO, PSO, EOA and SCA. In this scenario, the EGWO and GWO kept the same oscillation performance in different periods when the change of temperature was applied. The application of the change of temperature (period of 1.5 s and 2 s) for the PSO, SCA and EOA algorithms shows a power oscillation equal to 1.919 W, 2.035 W, and 2.035 W, respectively. Otherwise, at the period of 2.5 s and 3.5 s, the PSO, SCA and EOA algorithms show a power oscillation equal to 4.795 W, 2.517 W and 3.361 W, respectively. On the other hand, at the period of 4 s and 5 s, the PSO, SCA and EOA algorithms show a power oscillation equal to 1.496 W, 5.448 W and 10.759 W, respectively.

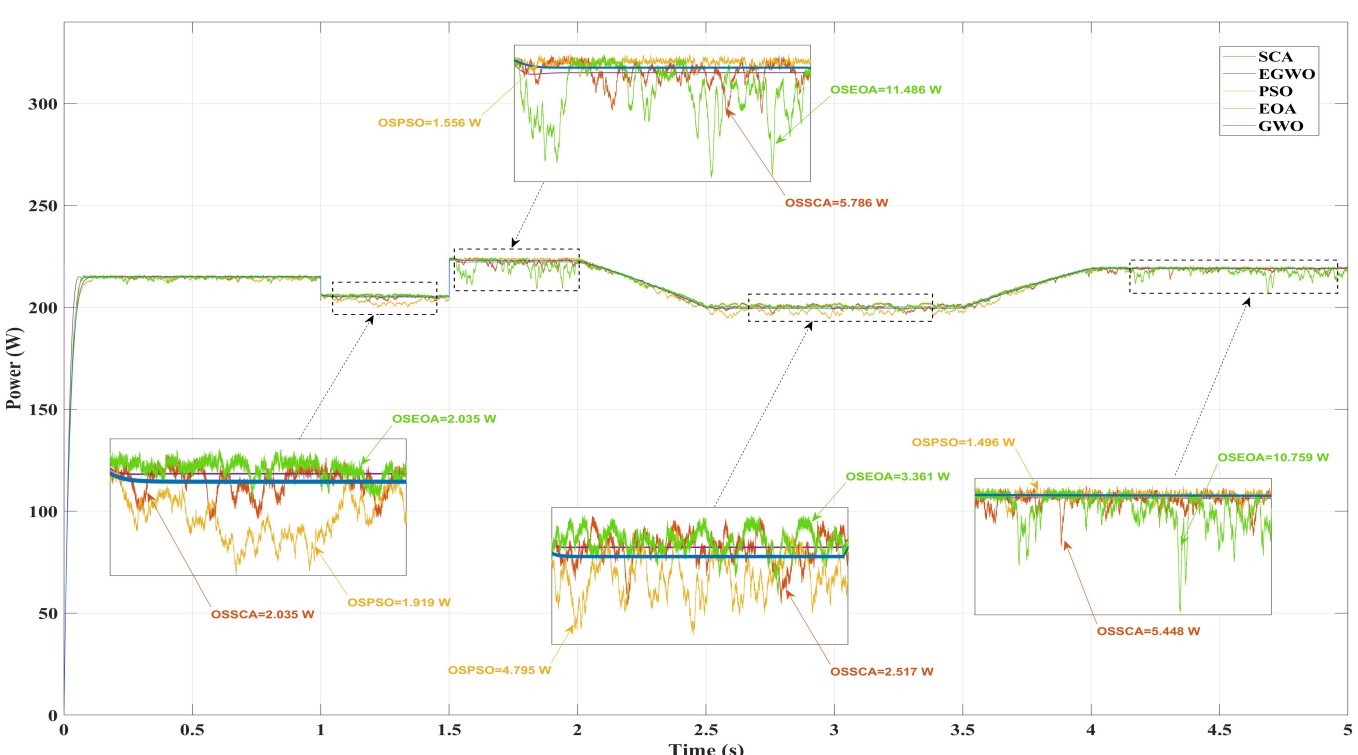

**Figure 8.** Photovoltaic power under variable temperature and fixed irradiance.

### 4.4. Fourth Scenario: Variable Temperature and Irradiance

Figure 9 represents the fourth scenario. In this scenario, variable temperature and irradiance are applied at the same periods.

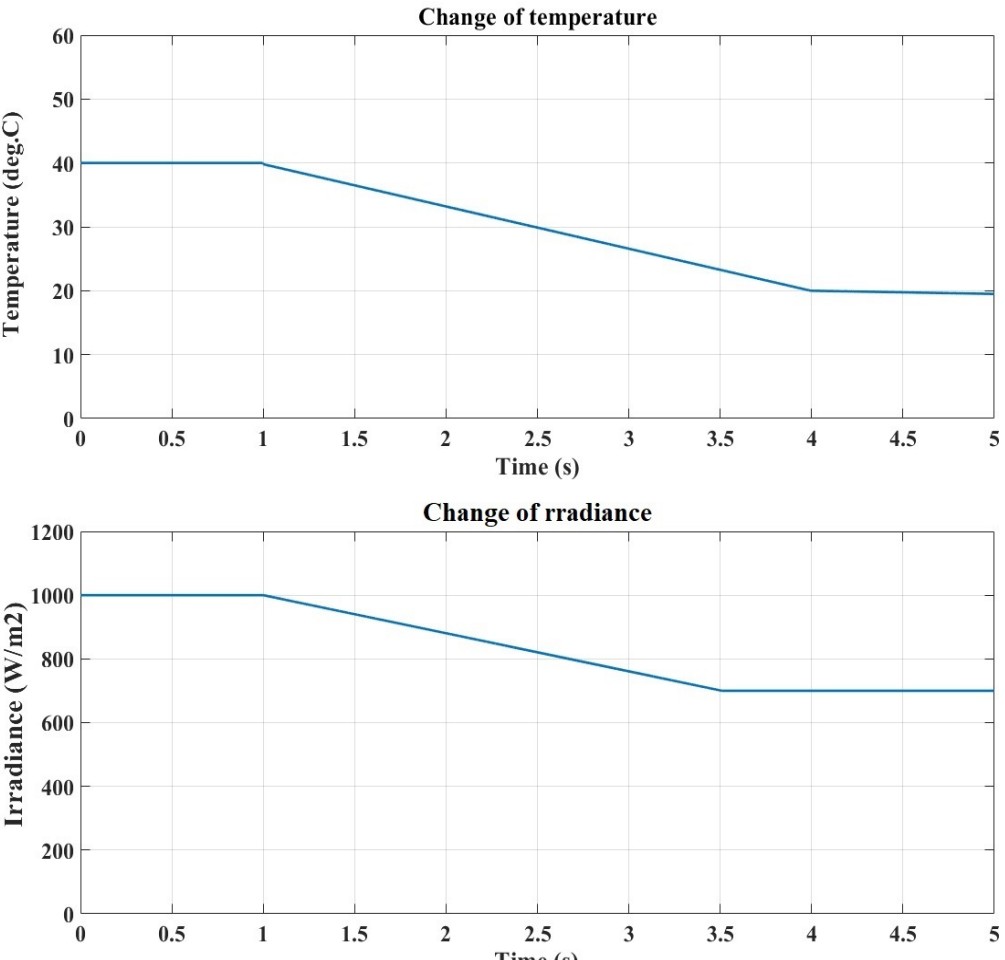

**Figure 9.** Variable irradiance and temperature.

Figure 10 shows the PV system power under EGWO, GWO, PSO, EOA and SCA. In this scenario, the EGWO and GWO again keep the same performance in different periods when the change of temperature and irradiance are applied. The application of the change of temperature and irradiance (period of 1 s and 5 s) in the PSO, SCA and EOA algorithms shows a power oscillation equal to 37.901 W, 35.379 W, and 39.323 W, respectively. According to these results, the increased oscillation observed in the PSO, SCA and EOA algorithms when there is a change in irradiance and temperatures is attributed to their slower response to sudden environmental changes. Hence, these algorithms take some time to adapt to new conditions, causing power oscillations as they seek the optimal operating point. Factors like algorithm complexity, parameter tuning, and transient environmental conditions can all contribute to this behavior. To mitigate such oscillations in dynamic environments, careful parameter tuning and consideration of algorithm design are essential.

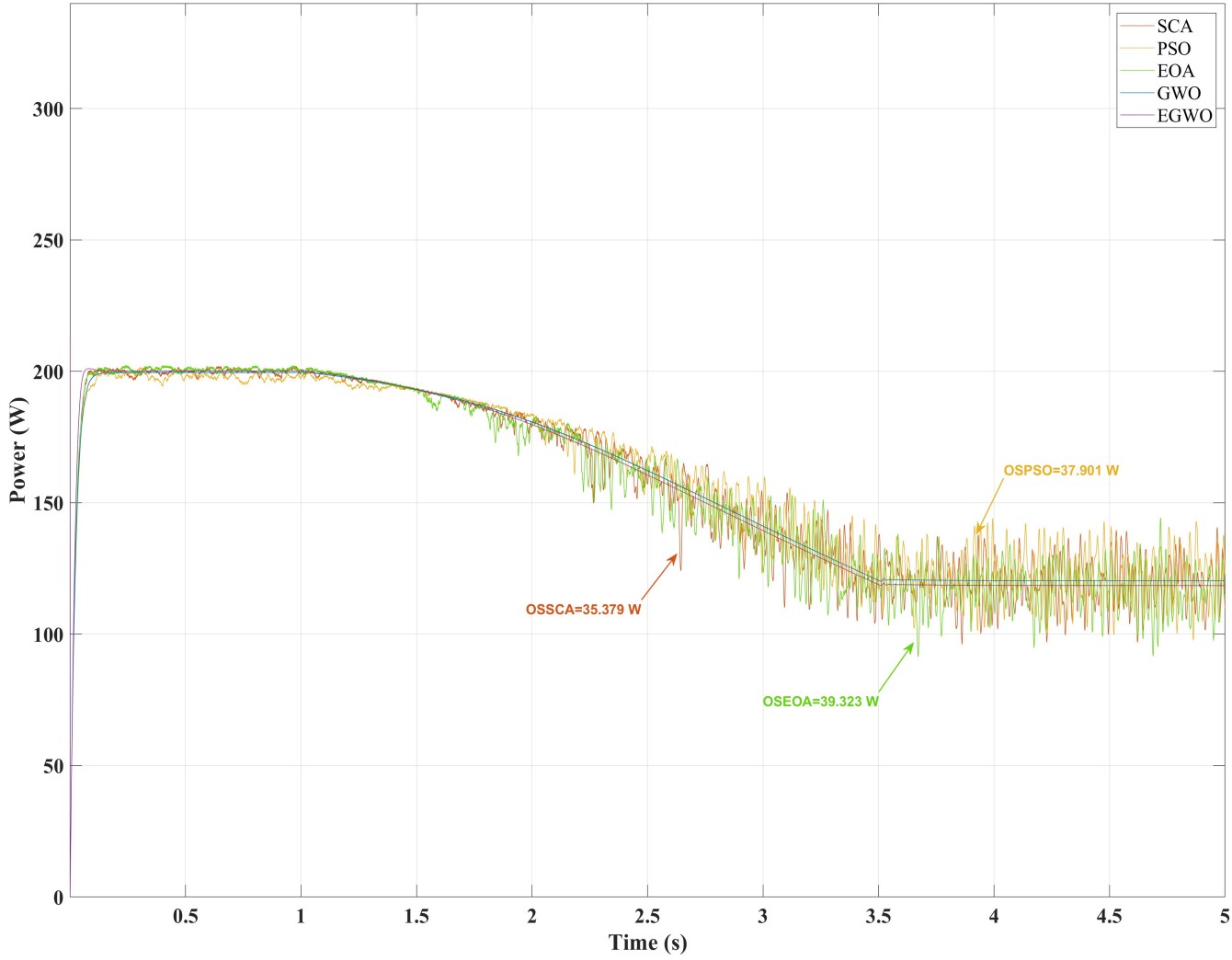

**Figure 10.** Photovoltaic power under variable irradiance and temperature.

In summary, the performance of the various MPPT algorithms was assessed under dynamic environmental conditions across multiple scenarios. In the first scenario, wherein solar irradiance was maintained at a constant value of 1000 W/m$^2$, and cell temperature was held at 25 °C, the superior performance of the EGWO algorithm was observed, characterized by rapid response and minimal power fluctuations. The GWO was identified as the second-best performer, as it demonstrated the ability to maintain a stable power output near the MPPT point. Conversely, other algorithms, such as the PSO, EOA and SCA, exhibited power oscillations, highlighting the potential for efficiency losses and reduced energy yield. In the second scenario, which introduced simultaneous changes in irradiance and temperature, heightened oscillations were observed in the PSO, SCA and EOA algorithms, owing to their complex responses to dynamic environmental conditions and interactions between variables. In the third and fourth scenarios, involving temperature variations, EGWO and GWO consistently maintained their performance, while varying levels of power oscillations were observed in the PSO, SCA and EOA algorithms. These findings underscore the importance of selecting appropriate MPPT algorithms tailored to specific environmental scenarios to minimize oscillations and optimize the overall efficiency of PV systems. Moreover, scalability plays a vital role in extending the application of these techniques to diverse system sizes and configurations. As we delve into the design and application of the proposed EGWO MPPT technique, it is essential to recognize that scalability is a critical aspect to be considered. The ability to optimally coordinate and share resources in large-scale PV systems is paramount, especially in the evolving energy landscape. The EGWO-based MPPT approach is inherently designed

for scalability, making it a versatile solution that efficiently adapts to systems of varying scales, such as small residential PV installations or a large utility-scale solar farm. The performance comparison of MPPT algorithms in different scenarios in terms of power fluctuations is presented in Table 4.

**Table 4.** Performance comparison of maximum power point tracking algorithms in different scenarios.

| Algorithm | Scenario 1 (Oscillation) | Scenario 2 (Oscillation) | Scenario 3 (Oscillation) | Scenario 4 (Oscillation) |
|---|---|---|---|---|
| EGWO | Excellent (0.09 W) | Excellent (0.09 W) | Excellent (0.09 W) | Excellent (0.09 W) |
| GWO | Good (0.266 W) | Good (0.266 W) | Good (0.266 W) | Good (0.266 W) |
| PSO | Poor (0.731 W) | Poor (18.401 W) | Poor (1.919 W) | Poor (37.901 W) |
| EOA | Poor (1.044 W) | Poor (26.671 W) | Poor (2.035 W) | Poor (39.323 W) |
| SCA | Poor (0.729 W) | Poor (15.796 W) | Poor (2.035 W) | Poor (35.379 W) |

## 5. Conclusions

In conclusion, this study provides valuable insights into the performance of MPPT algorithms in PV systems integrated with DC/DC converters. The results reveal the strengths and weaknesses of various MPPT techniques under diverse environmental scenarios. Notably, the EGWO stands out as a top-performing algorithm, consistently exhibiting swift response times and minimal power fluctuations around the MPP. This robust performance positions EGWO as a promising choice for real-world applications subject to variable environmental conditions. The GWO also demonstrates commendable stability and efficiency, making it a reliable alternative. However, this study underscores the limitations of other techniques, like PSO, EOA and SCA, which exhibit undesirable power oscillations under dynamic conditions, potentially leading to efficiency losses and reduced energy yield in PV systems. Therefore, careful selection of an appropriate MPPT algorithm tailored to specific operational requirements and environmental variability is of paramount importance.

Looking forward, this research offers several directions for future investigations. Researchers can delve into the integration of advanced techniques to further enhance MPPT adaptability in response to dynamic environmental changes. Exploring the performance of these algorithms across various renewable energy sources can provide a comprehensive understanding of their applicability. Additionally, in the managerial realm, the study highlights the significance of algorithm selection in optimizing the performance of PV systems, which can reduce operational costs and enhance energy conversion efficiency.

Nevertheless, it is important to recognize the limitations of this study. The performance evaluation primarily relies on simulations, and real-world applications may introduce additional complexities. Therefore, field studies and experimental validation are crucial to bridge the gap between simulation results and practical implementations. Moreover, this study predominantly focuses on a specific type of PV module, and it is important to consider the diverse responses of various PV technologies to MPPT algorithms.

In summary, this research not only advances our understanding of MPPT algorithms but also provides valuable insights for future research directions and managerial considerations within the renewable energy industry. By addressing these avenues and acknowledging the study's limitations, we can further enhance the efficiency and practical applicability of PV systems.

**Author Contributions:** Conceptualization, M.Y.S.; methodology, M.Y.S.; software, M.Y.S.; validation, M.Y.S., A.B. and O.B.; formal analysis, M.Y.S. and O.B.; investigation, M.Y.S., A.B., A.R. and O.B.; resources, O.B.; data curation, M.Y.S. and O.B.; writing—original draft preparation, M.Y.S.; writing—review and editing, M.Y.S., A.B., A.R. and O.B.; visualization, M.Y.S., O.B. and A.B.; supervision, O.B.; project administration, O.B.; funding acquisition, O.B. All authors have read and agreed to the published version of the manuscript.

**Funding:** The authors wish to express their gratitude to the Basque Government, through the project EKOHEGAZ II (ELKARTEK KK-2023/00051); to the Diputación Foral de Álava (DFA), through the project CONAVANTER; and to the UPV/EHU, through the project GIU20/063 for supporting this work.

**Institutional Review Board Statement:** Not applicable.

**Informed Consent Statement:** Not applicable.

**Data Availability Statement:** Data are contained within the article.

**Acknowledgments:** The authors wish to express their gratitude to the Basque Government, through the project EKOHEGAZ II (ELKARTEK KK-2023/00051); to the Diputación Foral de Álava (DFA), through the project CONAVANTER; and to the UPV/EHU, through the project GIU20/063, for supporting this work. Furthermore, the authors would like to acknowledge the projects supported by the Telecommunications Signals and Systems Laboratory (TSS), University Amar Telidji, which played a vital role in making this research possible. Additionally, sincere appreciation is extended to all the associates who have directly or indirectly contributed to this work.

**Conflicts of Interest:** The authors declare no conflict of interest.

## Abbreviations

The following abbreviations and nomenclatures are used in this manuscript:

| | |
|---|---|
| MPPT | Maximum power point tracking |
| MPP | Maximum power point |
| EGWO | Extended grey wolf optimizer |
| GWO | Grey wolf optimizer |
| EOA | Equilibrium optimization algorithm |
| PSO | Particle swarm optimization |
| SCA | Sin cos algorithm |
| P&O | Perturb and observe |
| INC | Incremental conductance |
| FOCV | Fractional open circuit voltage |
| FSCC | Fractional short circuit current |
| HL | Hill climbing |
| ABC | Artificial bee colony algorithm |
| GA | Genetic algorithm |
| ACO | Colony optimization |
| FA | Firefly algorithm |
| WOA | Whale optimization algorithm |
| CS | Cuckoo search |
| AFSA | Artificial fish swarm algorithm |
| PWM | Pulse width modulation |
| $d$ | Duty cycle |
| $P$ | PV system power (W) |
| $I_{out}$ | Cell current |
| $I_{ph}$ | Current generated by light |
| $I_d$ | Diode's current |
| $I_{sh}$ | Current of the parallel resistance |
| $I_0$ | Reverse saturation current |
| $V$ | Voltage across the PV cell |
| $V_d$ | Voltage of the equivalent diode |
| $R_s$ | Series resistance |
| $R_{sh}$ | Parallel resistance |
| $G$ | Actual irradiance |
| $G_{STC}$ | Irradiance at standard rating conditions |
| $K$ | Boltzmann constant |

| | |
|---|---|
| *q* | Electron charge |
| $T_c$ | Cell temperature |
| $T_r$ | Reference temperature |
| *n* | Diode ideality factor |
| $N_s$ | Number of series cells |

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
