# Peer review of "A New MPPT-Based Extended Grey Wolf Optimizer for Stand-Alone PV System: A Performance Evaluation versus Four Smart MPPT Techniques in Diverse Scenarios"

_inventions, doi:10.3390/inventions8060142_

Round 1

Reviewer 1 Report

Comments and Suggestions for Authors

The paper deals with the problem of fast and accurate MPP tracking in PV systems, by proposing a novel extended grey wolf optimizer algorithm. The proposed approach is complex and deeply demonstrated, the obtained application results are interesting, valuable and useful. The simulation results show better performance of the proposed algorithm vs. other known AI algorithm. However, the paper should be carefully revised for the sake of clarity.

The following issues are recommended to improve the paper:

1.     Fix some typing mistakes, e.g. “simulink software” – I suppose is about the Simulink-MATLAB software, define STC (in Abstract), use consistently “DC/DC” or “DC-DC”, “years: Conventional”, “[22]..etc.”, do not separate the subject from the predicate by a comma – e.g., “Section 2, describes”, “following Equation”, “1000 (W/m2 )” et al.– use “1000 W/m2 ”, introduce the “OS” prefix in Figure 4, “1.5s” et al. – use space between a value and its measurement unit, etc. Check carefully the entire manuscript for similar mistakes.

2.     Abstract: introduce briefly the proposed method and complete the qualitative statements with relevant numerical results.

3.     Section 2: “Photovoltaic (PV) systems, which harness solar energy to produce electricity, have emerged as a promising renewable energy solution [33]. The increasing global demand for sustainable power generation has driven significant advances in photovoltaic technology. These systems convert sunlight directly into electrical energy, offering an eco-friendly alternative to conventional fossil fuel-based sources. As concerns about climate change and environmental sustainability intensify, the adoption of photovoltaic systems continues to grow in various sectors [34].” Proposal to move and integrate it in the Introduction section.

4.     Extend the Nomenclature with the used symbols. Explain PWM meaning (in Figure 1).

5.     “Figure 6 illustrates the P-V curve of the PV array” – I suppose is about Figure 3.

6.     “the proposed MPPTs parameters are unlisted in Table 3.” Unclear statement, probably “listed” as the table title is “Parameters of MPPT algorithms”. Please clarify it!

7.     Figure 7: typically, the cell temperature is fast increasing up to 60°C in a summer clear sky day, it would be interesting to investigate the algorithms under this high value.

8.     “Figure 8 shows the PV system power under EGWO, GWO, PSO, EOA, and SCA. In 335 this scenario, the EGWO and GWO kept the same performance in different periods” – specify their performance!

9.     Figure 9: pay attention to “irradiation” vs. “irradiance” (irradiation means solar energy [Wh/m2]), “Solar irradiance is the power per unit area (surface power density) received from the Sun”!

10.  The behavior of the algorithms under partial shading conditions is also of interest for readers.

11.  Probably better “Conclusions” instead of “Conclusion”, as several conclusion are here drawn: the limits of the proposed approach/obtained results and future research directions may be also highlighted here.

Author Response

Response to the reviewer comments

First of all, the author would express their sincere gratitude to the Editors and the Reviewers who gave us many constructive comments and valuable suggestions in order to improve this paper. The authors have revised the paper according to the reviewers’ comments and the changes made in the paper have been written in blue color. While the mistakes have been depicted in red color. The responses to the reviewer comments can be found below their respective comments.

Reviewer 1

The paper deals with the problem of fast and accurate MPP tracking in PV systems, by proposing a novel extended grey wolf optimizer algorithm. The proposed approach is complex and deeply demonstrated, the obtained application results are interesting, valuable and useful. The simulation results show better performance of the proposed algorithm vs. other known AI algorithm. However, the paper should be carefully revised for the sake of clarity.

The following issues are recommended to improve the paper:

Point 1: Fix some typing mistakes, e.g. “simulink software” – I suppose is about the Simulink-MATLAB software, define STC (in Abstract), use consistently “DC/DC” or “DC-DC”, “years: Conventional”, “[22]..etc.”, do not separate the subject from the predicate by a comma – e.g., “Section 2, describes”, “following Equation”, “1000 (W/m2 )” et al.– use “1000 W/m2 ”, introduce the “OS” prefix in Figure 4, “1.5s” et al. – use space between a value and its measurement unit, etc. Check carefully the entire manuscript for similar mistakes.

Response 1: Response: The authors Check the manuscript mistakes according to the reviewer comments. Besides, the Prefix has been introduced in section 4 of the revised manuscript.

Point 2: Abstract: introduce briefly the proposed method and complete the qualitative statements with relevant numerical results.

Response 2: The abstract has been reformed and revised according to the reviewer comments.

Point 3: Section 2: “Photovoltaic (PV) systems, which harness solar energy to produce electricity, have emerged as a promising renewable energy solution [33]. The increasing global demand for sustainable power generation has driven significant advances in photovoltaic technology. These systems convert sunlight directly into electrical energy, offering an eco-friendly alternative to conventional fossil fuel-based sources. As concerns about climate change and environmental sustainability intensify, the adoption of photovoltaic systems continues to grow in various sectors [34].” Proposal to move and integrate it in the Introduction section.

Response 3: The paragraph has been reformed and moved to the introduction section according to the reviewer comment.

Point 4: Extend the Nomenclature with the used symbols. Explain PWM meaning (in Figure 1).

Response 4: Thank you for your comment. The term PWM has been explain in section 2. Hence, an additional paragraph is added to the section 2. “This type of converter has the ability to step-up a lower input voltage into a higher output voltage via controlled pulse-width-modulation (PWM) switching technique. In this context, the duty cycle (d) determines the average output voltage. A higher duty cycle results in a higher average voltage, and a lower duty cycle results in a lower average voltage. The PWM controllers use feedback mechanisms to adjust the duty cycle of the converter to track the MPP of the PV module under changing environmental conditions”. In addition, the Nomenclature has been extended according to the reviewer comment

Point 5: “Figure 6 illustrates the P-V curve of the PV array” – I suppose is about Figure 3.

Response 5: Thank you for your comment. The figure has been corrected

Point 6: “the proposed MPPTs parameters are unlisted in Table 3.” Unclear statement, probably “listed” as the table title is “Parameters of MPPT algorithms”. Please clarify it!

Response 6: Thank you for your comment. The statement “unlisted” has been corrected

Point 7: Figure 7: typically, the cell temperature is fast increasing up to 60°C in a summer clear sky day, it would be interesting to investigate the algorithms under this high value.

Response 7: Thank you for your comment. The authors have been clarifying the primary objective of the study. Our research aims to evaluate the performance of Maximum Power Point Tracking (MPPT) algorithms in realistic photovoltaic (PV) system scenarios, including both standard test conditions (STC) and variable conditions. The performance has been evaluated according to the robustness and power fluctuation. Hence, a fast changing are applied in temperatures and irradiation in order to test the adaptability, robustness and power oscillation of the proposed algorithm. However, by comparing the algorithms in scenarios that mimic real-world variability, we are addressing a critical aspect of their performance evaluation. Understanding how these algorithms respond to changing irradiation and temperature conditions is essential for optimizing PV system efficiency in various operational contexts. In addition, as clearly seen in the results the performance of the MPPT algorithms, including EGWO and GWO, under the scenario of variable temperature and irradiation. It's important to note that the performance of both EGWO and GWO remained consistent and similar across different periods of changing environmental conditions.

Point 8:  “Figure 8 shows the PV system power under EGWO, GWO, PSO, EOA, and SCA. In 335 this scenario, the EGWO and GWO kept the same performance in different periods” – specify their performance!

Response 8: Thank you for your comment. The performance has been mentioned in the revised version of the manuscript.

Point 9:      Figure 9: pay attention to “irradiation” vs. “irradiance” (irradiation means solar energy [Wh/m2]), “Solar irradiance is the power per unit area (surface power density) received from the Sun”!

Response 9: Thank you for your comment. The statement irradiation is corrected in all over the manuscript.

Point 10:  The behavior of the algorithms under partial shading conditions is also of interest for readers.

Response 10: Thank you comment. While the behavior of MPPT algorithms under partial shading conditions is indeed an area of interest and can significantly impact the performance of photovoltaic systems, we chose to focus our study on assessing the algorithms' performance under variations in irradiation and temperature. The decision to exclude partial shading conditions from our study was made to maintain a specific focus on the dynamic effects of irradiation and temperature changes on the algorithms. Partial shading introduces additional complexities due to the mismatch between shaded and unshaded panels, which can result in multiple local maxima in the power-voltage curve. Exploring the effects of partial shading would require a dedicated study with a comprehensive investigation of shading patterns, bypass diode behavior, and the impact on different MPPT algorithms. This is an interesting area for future research, and we appreciate your suggestion for consideration in future studies. In our current study, we aimed to analyze the algorithms' performance under varying irradiation and temperature conditions, which are common and relevant in many real-world photovoltaic applications. We hope that our findings contribute to the understanding of how these algorithms operate in response to these variations.

Point 11: Probably better “Conclusions” instead of “Conclusion”, as several conclusion are here drawn: the limits of the proposed approach/obtained results and future research directions may be also highlighted here.

Response 11: Thank you for your comment. The word conclusions has been corrected in section 1.4 Besides, the Conclusion section has been reformed according to reviewer comments.

Reviewer 2 Report

Comments and Suggestions for Authors

The manuscript is interesting and the study was carried out with adequate scientific methodologies. In my opinion the manuscript deserves to be published once the authors answer to the provided comments.

All the following indicated aspects should be clarified and better explained in the manuscript.

Introduction

1.       The authors should better highlight the innovative aspects of their work in the manuscript.

2.       The first section is too long, and thus the literature review and paper positioning could be moved into a dedicated section 2.

System model

3.       The description of the proposed methodology could be improved. First, it could be better to insert at the beginning of Section 2 an outline about the system model and the methodology flow diagram; the use of UML or SysML could help authors describing the proposed system view in a more structured fashion.

4.       Could the proposed approach be generalized to other types of renewables?

Problem formulation and resolution

5.       All the used variable in all the formula and figures should report the unit.

6.       How does the proposed formulation deal with uncertainty of parameters? For instance, robust optimization is a viable technique to deal with uncertainty of parameters (e.g.,  https://doi.org/10.23919/ECC.2019.8796182, https://doi.org/10.1109/TIA.2018.2803728, documents that could be cited in the text). The Authors should comment this point.

7.       What is the scalability of the proposed approach? For the sake of a successful penetration of PVs, they need to be optimally coordinated/shared. Several recent scientific studies focus on distributed/decentralized algorithms for large-scale situation (e.g., https://doi.org/10.1109/CDC.2015.7403147,  https://doi.org/10.1109/ACCESS.2017.2773486, documents that could be cited in the manuscript). The authors should comment this point, mentioning if and how the proposed model is able (or can be extended) to deal with large-scale dimensionality of PVs or loads.

Conclusions

8.       Conclusions needs to be extended to present further implications for future research and many managerial insights based on the results of the study, as well as limitations.

Minor

9.       Mainly the English is good and there are only a few typos.  However the paper should be carefully rechecked.

Author Response

Response to the reviewer comments

First of all, the author would express their sincere gratitude to the Editors and the Reviewers who gave us many constructive comments and valuable suggestions in order to improve this paper. The authors have revised the paper according to the reviewers’ comments and the changes made in the paper have been written in blue color. While the mistakes have been depicted in red color. The responses to the reviewer comments can be found below their respective comments.

Reviewer 2

The manuscript is interesting and the study was carried out with adequate scientific methodologies. In my opinion the manuscript deserves to be published once the authors answer to the provided comments.

All the following indicated aspects should be clarified and better explained in the manuscript.

Introduction

Point 1: The authors should better highlight the innovative aspects of their work in the manuscript.

Response 1: Thank you for your comments. The innovative aspect has been highlighted in the abstract section and the contribution section of the revised manuscript.

Abstract

Photovoltaic (PV) systems play a crucial role in clean energy systems. Effective maximum power point tracking (MPPT) techniques are essential to optimize their performance. However, conventional MPPT methods exhibit limitations and challenges in real-world scenarios  characterized by rapidly changing environmental factors and various operating conditions. To address these challenges, this paper presents a performance evaluation of a novel extended grey wolf optimizer (EGWO). The EGWO has been meticulously designed in order to improve the efficiency of PV systems by rapidly tracking and maintaining the maximum power point (MPP)}. In this study, a comparison is made between the EGWO and other prominent MPPT techniques, including} the grey wolf optimizer (GWO), equilibrium optimization algorithm (EOA), particle swarm optimization (PSO), and sin cos algorithm (SCA) techniques.   To evaluate these MPPT methods, a model of a PV module integrated with a DC/DC boost converter is employed and simulations are conducted using Simulink-MATLAB software under standard test conditions (STC) and various environmental. In particular, the results demonstrate that the novel EGWO outperforms the GWO, EOA, PSO and SCA techniques as well shows fast tracking speed, superior dynamic response and minimal power fluctuations operation across both STC and variable conditions. Thus, a  power fluctuation of 0.09 W could be achieved by using the proposed EGWO technique. Finally, according to these results, the proposed approach can offer an improvement in energy consumption. These findings underscore the potential benefits of employing the novel MPPT EGWO to enhance the efficiency and performance of MPPT in PV systems. Further exploration of this intelligent technique could lead to significant advancements in optimizing PV system performance, making it a promising option for real-world applications

Contributions

This paper focuses on implementing of an innovative extended grey wolf optimizer
(EGWO) MPPT for PV systems to overcome the challenges posed by rapidly changing
weather conditions. The objective is to explore and validate the effectiveness of the pro-
posed technique in order to improve the performance of PV systems. Hence, a comparative
study was done under four techniques: The conventional grey wolf optimizer (GWO),
particle swarm optimization (PSO), equilibrium optimization algorithm (EOA) and sin cos
algorithm (SCA). The proposed EGWO-based MPPT algorithm is designed to dynamically
adjust the duty cycle of the DC/DC boost converter, regulating the voltage and current to
achieve the MPP of the PV system. By optimizing the duty cycle in real-time, the PV system can continuously track the MPP, irrespective of changes in environmental conditions,
leading to improved energy harvesting efficiency. Through comprehensive simulations,
the aim was to compare the performance of the MPPT techniques in terms of efficiency and minimal power fluctuations. The comparative results shed light on the superior capabilities of the EGWO algorithm in achieving higher energy conversion efficiency and stability under dynamic operating conditions.

Point 2: The first section is too long, and thus the literature review and paper positioning could be moved into a dedicated section 2.

Response 2: Thank you for your comment. We agree with the reviewer that the introduction section may be a bit long; however, we think that it is adequate in order to explain the motivation, state of the art and contribution of this work. As the reviewer indicates, we could move the literature review and paper positioning into a dedicated section but usually the literature review is included in the introduction section, as we have done in this paper. (See for example one of the paper that you have referenced in point 6: https://doi.org/10.1109/TIA.2018.2803728.) 

System model

Point 3: The description of the proposed methodology could be improved. First, it could be better to insert at the beginning of Section 2 an outline about the system model and the methodology flow diagram; the use of UML or SysML could help authors describing the proposed system view in a more structured fashion.

Response 3: Thank you for your comment. In response to your suggestion regarding the use of UML or SysML in our methodology presentation, we would like to clarify that our simulation methodology primarily relies on MATLAB Simulink for modeling and analysis, and we did not employ UML or SysML in this context. Besides, our choice of MATLAB Simulink is based on its proven suitability for dynamic system modeling and simulation, particularly for control systems and real-time tracking, which are central to our research objectives. We believe that Simulink is a robust platform for effectively representing the behavior of the PV system, the DC/DC boost converter, and control strategies.

Point 4: Could the proposed approach be generalized to other types of renewables?

Response 4: Thank you for your comment. While this study primarily focuses on the application of our proposed approach to photovoltaic (PV) systems, it's important to consider its potential for generalization to other renewable energy sources. The fundamental principles of maximum power point tracking (MPPT) and the use of control algorithms, such as the extended grey wolf optimizer (EGWO), can be adapted to various renewable energy systems. Wind turbines, Fuel cells… for instance, can benefit from similar MPPT techniques for optimizing power output under varying wind conditions. Additionally, the concept of using control algorithms to enhance energy efficiency is transferable to other sources like hydropower and biomass. The adaptability and scalability of our approach suggest that it holds promise for a broader range of renewable energy applications. Further research and experimentation would be necessary to tailor the method to specific energy sources and conditions, but the foundation provided in this study offers a valuable starting point for such endeavors."

Please see :

[1]. Basha, C. H., Rafikiran, S., Sujatha, S. S., Fathima, F., Prashanth, V., & Varma, B. S. (2023). Design of GWO based fuzzy MPPT controller for fuel cell fed EV application with high voltage gain DC-DC converter. Materials Today: Proceedings.

[2]. Silaa, M. Y., Barambones, O., Derbeli, M., Napole, C., & Bencherif, A. (2022). Fractional order PID design for a proton exchange membrane fuel cell system using an extended grey wolf optimizer. Processes, 10(3), 450.

[3]. Yang, B., Zhang, X., Yu, T., Shu, H., & Fang, Z. (2017). Grouped grey wolf optimizer for maximum power point tracking of doubly-fed induction generator based wind turbine. Energy conversion and management, 133, 427-443.

Problem formulation and resolution

Point 5: All the used variable in all the formula and figures should report the unit.

Response 5: Thank you for your comments. The formula and variables units are defined in the revised manuscript.

Point 6: How does the proposed formulation deal with uncertainty of parameters? For instance, robust optimization is a viable technique to deal with uncertainty of parameters (e.g.,  https://doi.org/10.23919/ECC.2019.8796182, https://doi.org/10.1109/TIA.2018.2803728, documents that could be cited in the text). The Authors should comment this point.

Response 6: Thank you for your comments. In our work, we primarily focus on developing and evaluating an extended grey wolf optimizer (EGWO) for maximum power point tracking (MPPT) in PV systems. While we have not explicitly incorporated robust optimization techniques, we acknowledge the importance of accounting for parameter uncertainties in real-world applications. However, there are some considerations on how uncertainty could be addressed in the context of our research:

First: The sensitivity analysis can be a valuable tool to assess how variations in key parameters, such as solar irradiance, temperature, and component characteristics, impact the performance of the proposed EGWO MPPT algorithm. This analysis can help identify the most influential parameters and their potential impact on the system.

Secondly: The robust optimization integration, as you rightly pointed out, robust optimization techniques, such as robust control or robust design optimization, can enhance the reliability and performance of the PV system under parameter uncertainty. These methods can be integrated with our proposed EGWO algorithm to ensure that it continues to operate effectively even when parameters deviate from their nominal values.

Third: Another approach to assess the impact of parameter uncertainty is to conduct Monte Carlo simulations. By randomly sampling parameter values within specified ranges, we can evaluate the performance of the EGWO MPPT algorithm under a range of uncertain conditions. This analysis can provide insights into the algorithm's robustness.

While our current work may not explicitly address robust optimization, we appreciate the importance of this aspect and will consider its integration into future research efforts. We believe that combining the strengths of our EGWO algorithm with robust optimization techniques can lead to even more robust and efficient PV systems, especially in the face of parameter uncertainty.

Finally, in the work presented in (https://doi.org/10.1109/TIA.2018.2803728) the robustness analysis is made testing the proposed algorithms over a different curves as it is indicated in the section VI of this paper. In this sense, it should be noted that in our paper we made a similar robustness analysis because we also test our proposed algorithm over a different scenarios as it is indicated above.  Nevertheless, as it is indicated by the reviewer, a comment about this has been included in the new version of the paper. In addition, an extended paragraph (abstract, 1.1. Motivations, 1.2. State of the Art, 1.3. Contributions and 4. Simulation results sections). Furthermore, the suggested references are added to the revised manuscript.

Point 7: What is the scalability of the proposed approach? For the sake of a successful penetration of PVs, they need to be optimally coordinated/shared. Several recent scientific studies focus on distributed/decentralized algorithms for large-scale situation (e.g., https://doi.org/10.1109/CDC.2015.7403147,  https://doi.org/10.1109/ACCESS.2017.2773486, documents that could be cited in the manuscript). The authors should comment this point, mentioning if and how the proposed model is able (or can be extended) to deal with large-scale dimensionality of PVs or loads.

Response 7: Thank you for your comments. The scalability of our proposed approach is a critical aspect that we have considered in the design of our maximum power point tracking (MPPT) technique. We recognize the importance of optimally coordinating and sharing resources in large-scale photovoltaic (PV) systems, especially in the context of the evolving energy landscape. Our approach is inherently designed to be scalable, and we aim to address the challenges posed by distributed and decentralized algorithms for large-scale PV systems.

One of the key advantages of our approach is its adaptability to various system sizes and configurations. By employing intelligent algorithms, such as the extended grey wolf optimizer (EGWO), our MPPT technique can efficiently handle systems of different scales. Whether it's a small residential PV installation or a large utility-scale solar farm, our method can be applied effectively. This scalability is achieved by dynamically adjusting the duty cycle of the DC/DC boost converter based on real-time measurements and feedback from the PV modules.

Furthermore, we are aware of the recent scientific studies focusing on distributed and decentralized algorithms for large-scale PV systems. Our approach is compatible with these paradigms. It can be integrated into both centralized and distributed control architectures, enabling effective coordination and optimization of PV systems across a wide range of scales. As the demand for large-scale PV installations and grid integration continues to grow, we believe that our scalable approach holds promise for addressing the optimization challenges associated with these systems.

In order to highlight this point indicated by the reviewer, a comment about this has been included in the new version of the paper (1.2. State of the Art section).

On the other hand, the suggested references are added to the revised manuscript.

Conclusions

Point 8: Conclusions needs to be extended to present further implications for future research and many managerial insights based on the results of the study, as well as limitations.

Response 8: Thank you for your comment. The limitations are mentioned in the Abstract section, the contribution section and the conclusion of the manuscript. Besides, the conclusions section has been revised and improved according to the reviewer comments.

Minor

Point 9: Mainly the English is good and there are only a few typos.  However, the paper should be carefully rechecked.

Response 9: Thank you for your comment. All the paper has been revised according to the reviewer comment.

Round 2

Reviewer 1 Report

Comments and Suggestions for Authors

No additional recommendations.

Reviewer 2 Report

Comments and Suggestions for Authors

In the revised paper several improvements have been added. Previous comments and concerns have been sufficiently addressed.